# The Impact of Non-Invasive Ventilation on Sleep Quality in COPD Patients

**DOI:** 10.3390/jcm11185483

**Published:** 2022-09-19

**Authors:** Maximilian Wollsching-Strobel, Iris Anna Bauer, Johannes Julian Baur, Daniel Sebastian Majorski, Friederike Sophie Magnet, Jan Hendrik Storre, Wolfram Windisch, Sarah Bettina Schwarz

**Affiliations:** 1Department of Pneumology, Cologne Merheim Hospital, Kliniken der Stadt Köln, 51109 Cologne, Germany; 2Faculty of Health, School of Medicine, Witten/Herdecke University, 58455 Witten, Germany; 3Department of Pneumology, University Medical Hospital, 79110 Freiburg im Breisgau, Germany; 4Praxis Pneumologie Solln, 81477 Munich, Germany

**Keywords:** sleep quality, COPD, non-invasive ventilation, polysomnography, chronic respiratory insufficiency

## Abstract

Background: Non-invasive ventilation (NIV) has been shown to be the most appropriate therapy for COPD patients with chronic respiratory failure. While physiological parameters and long-term outcome frequently serve as primary outcomes, very few studies have primarily addressed the impact of NIV initiation on sleep quality in COPD. Methods: This single-center prospective cohort study comprised NIV-naïve patients with COPD. All patients underwent polysomnographic evaluation both at baseline and at 3 months follow-up, accompanied by the assessment of health-related quality of life (HRQL) using the Severe Respiratory Insufficiency Questionnaire (SRI) and the Epworth Sleepiness Scale (ESS). A subgroup evaluation was performed to address the impact of comorbid obstructive sleep apnea syndrome (OSAS). Results: Forty-six patients were enrolled and twenty-five patients completed the follow-up period (66.7 ± 7.4 years). NIV resulted in an increase in slow-wave sleep (+2% (−3.5/7.5), *p* = 0.465) and rapid eye movement sleep (+2.2% (−1.0/5.4), *p* = 0.174), although no statistical significance could be detected. ESS (−1.7(−3.6/0.1), *p* = 0.066) also showed a positive trend. Significant improvements in the Respiratory Disturbance Index (RDI) (−12.6(−23.7/−1.5), *p* = 0.027), lung function parameters, transcutaneous PCO_2_ and the SRI summary scale (4.5(0.9/8), *p* = 0.016) were observed. Conclusion: NIV therapy does not decrease sleep quality and is even capable of improving HRQL, transcutaneous PaCO_2_, daytime sleepiness and RDI, and the latter especially holds true for patients with comorbid OSAS.

## 1. Introduction

Non-invasive mechanical ventilation (NIV) is an established therapeutic option for chronic respiratory failure across a variety of etiologies, including COPD, neuromuscular diseases (NMDs), obesity hypoventilation syndrome (OHS) and chest wall disorders [1]. Patients with chronic hypercapnic respiratory failure due to COPD are not only associated with the largest increase in NIV use, but also form in many European countries the largest proportion of patients on NIV [2]. NIV is aimed at restoring the balance between load and capacity in respiratory muscles, as well increasing alveolar ventilation. Indeed, when NIV is aimed at maximally reducing the partial pressure of carbon dioxide (PaCO_2_) in patients with chronic hypercapnic respiratory failure following acute exacerbation of COPD, improved survival and increased admission-free survival are evident. This form of therapy is defined as high-intensity non-invasive positive-pressure ventilation (HI-NIV) [3]. HI-NIV requires ventilator settings to be increased in a stepwise approach, either to an individually tolerated maximum, or to the levels needed for achieving normocapnia [4]. 

Although there is increasing evidence for the positive effects of NIV, certain scientific aspects remain unresolved. For this reason, the European Respiratory Society (ERS) taskforce described in their 2019 guidelines the research topics that need to be prioritized [5]. In particular, they noted that there was a current lack of scientific evidence for the impact of NIV on sleep quality, which has only rarely been used as a primary endpoint in the majority of NIV studies to date [6]. In addition, the treatment effects of NIV have not been adequately elucidated in patients with COPD and obstructive sleep apnea syndrome (OSAS) (also known as overlap syndrome), even though patients with comorbid OSAS have a higher mortality rate [7].

The ERS Task Force therefore suggested that research into the mutual influence of chronic respiratory insufficiency and OSAS on therapeutic outcome should receive priority [5]. Therefore, by applying methods such as polysomnography and assessing multiple patient-related outcomes, in the present study, we aimed to address the efficacy of NIV and its impact on sleep quality in COPD patients. The findings are intended to serve as a basis for further research regarding these important issues in a rapidly growing patient population, assuming that NIV has a beneficial effect on sleep quality.

## 2. Materials and Methods

The study protocol was approved by the Institutional Review Board for Human Studies at the Bavarian Medical Association in Germany, and performed in accordance with the ethical standards laid down in the Declaration of Helsinki (last revised in October 2013) [8]. Written informed consent was obtained from all subjects. The study was prospectively registered at the German Clinical Trials Register (DRKS00015496).

### 2.1. Patients

Adults with stable COPD were enrolled between August 2018 and January 2020 at the Department of Intensive Care, Sleep Medicine and Mechanical Ventilation, Asklepios Fachkliniken, Munich-Gauting. The diagnosis of COPD was based on criteria established by the GOLD report [9].

COPD patients were classified as stable if they had (i) no clinical signs of exacerbation 4 weeks prior to enrollment, and (ii) a pH > 7.35 [9]. All participants had respiratory insufficiency with an indication for the initiation of NIV based on German guidelines [10,11]. A detailed overview of the indication criteria can be found in Appendix A. Ventilation settings were aimed at normalizing PaCO_2_ levels or establishing the most tolerable level of therapy that would allow the greatest possible decrease in PaCO_2_ [12]. Patients with mental retardation, patients who were unable to give informed consent and patients with deafness were excluded from the study. In order to ensure the generalizability of the results, the remaining exclusion criteria were reduced to a minimum, in accordance with recommendations for pragmatic trials [13].

Two groups were defined for the subgroup analysis, and each differed in the presence of comorbid OSAS, which was defined by the detection of >10 events of obstructive breathing disorder during baseline polysomnography.

### 2.2. Study Design and Measurements

This prospective, single-center cohort study investigated sleep quality in patients with severe COPD before and after three months of NIV initiation therapy. 

The study procedure is shown in Figure 1. All participants underwent lung function assessment using spirometry, full-body plethysmography, respiratory muscle function testing (Fa. Jaeger, MasterScreen™ Body, Vyaire Medical, 97204 Hochberg, Germany) and capillary blood gas analyses from the arterialized earlobe (Radiometer GmbH, ABL 800 flex, 47807 Krefeld, Germany). Furthermore, each patient was monitored by polysomnography (PSG) (SOMNOscreenTM plus, SOMNOmedics GmbH, 97236 Randersacker, Firmware E09 08/09/16–Software Domino 2.9.0–26/2/18) to assess sleep quality prior to NIV initiation and after 3 months of home mechanical NIV. A qualified physician who was blinded to the study participation of the respective patients performed the data analysis according to guidelines by the American Academy of Sleep Medicine (AASM) 2017 and the German Society for Sleep Medicine (DGSM) 2017 [14,15]. Supplementary overnight transcutaneous PCO_2_ monitoring was performed (PtcCO_2_; SenTec DM^®^, Software V-STATS 4.0; SenTec AG; Therwil, Switzerland). Health-related quality of life (HRQL) was assessed using the Severe Respiratory Insufficiency Questionnaire (SRI). This questionnaire was developed for patients with severe respiratory failure. In the corresponding validation study, high psychometric values for reliability and validity were demonstrated for SRI, especially for COPD patients [16,17]. Moreover, it is the primary tool for measuring HRQL in patients requiring NIV, where a minimal, clinically relevant difference of 5 points along with a set of higher scores indicate better HRQL [16]. In addition, daytime sleepiness was assessed by the Epworth Sleepiness Scale (ESS) questionnaire, with lower scores indicating less daytime sleepiness [18].

### 2.3. Data Management and Statistical Analysis

All data were entered into a standardized case report form (CRF), subsequently documented in a pseudonymous fashion, and archived electronically. SRI scores were only calculated if the proportion of missing values was less than 10%.

The primary aim of this study was to compare rapid eye movement (REM) sleep time before and after the initiation of long-term NIV in patients with COPD. Since insufficient data are available on sleep quality in COPD patients undergoing NIV therapy, the study was conducted as a pilot study and no sample size calculation was performed. The minimum sample size for statistical analysis was set at 35 participants. The results were primarily analyzed in a descriptive fashion. For subgroup analyses, patients were divided into two groups according to the presence of comorbid OSAS, as previously described [15].

All statistical calculations were performed using SigmaPlot 12.3 (Systat Software GmbH, Erkrath, Germany) and SPSS Statistics 28 (IBM Corporation, New York, NY, USA). Comparison of sleep quality between groups was performed using the 2-tailed Student’s *t*-test for normally distributed data and the Mann–Whitney rank sum test for data with a non-normal distribution. Data values are presented as mean ± standard deviation (SD). Due to the small number of cases, the calculations primarily served as the basis for a descriptive analysis. Therefore, although *p*-values < 0.05 were considered significant, they do not actually represent statistical significance. Therefore, no correction was made for multiple testing. Cohan’s *d* was used to determine effect size, except in the case of SRI scores, where a minimum clinically relevant difference of ±5 points is established for COPD patients [19].

## 3. Results

A total of 50 patients were screened for eligibility. Four patients chose not to participate, and twenty-one follow-up visit measurements were excluded due to incomplete data or because patients had withdrawn their willingness to participate. Twenty-five patients were ultimately included in the final analysis (Figure 2). Demographic data as well as lung function parameters are displayed in Table 1. The mean IPAP value after the 3-month NIV period was 20 ± 3 cm H_2_O, with an EPAP of 7 ± 2 cm H_2_O and a back-up respiratory rate on NIV of 15 ± 2 breaths/min. Twenty-one patients (84%) received supplemental oxygen during NIV (2.2 ± 0.7 l).

The results of the polysomnographic measurements before and 3 months after the initiation of NIV therapy are shown in Figure 3. In terms of the primary endpoint, there was a mean increase of 2.2% in the proportion of REM sleep (95% CI −1.0/5.4; *p* = 0.174, Cohan’s d = 0.28 (−0.12/0.68)), although this did not reach significance. The polysomnographic results for each of the subgroups are demonstrated in the Appendix A. The mean Respiratory Disturbance Index (RDI) measured via PSG was 20.8 ± 27.1 per hour. Based on the previously determined cut-off of RDI ≥ 10, 15 patients were assigned to the subgroup without comorbid OSAS, and 10 patients to the subgroup with comorbid OSAS.

The results of the SRI measurements used to assess the impact of NIV therapy on disease-specific HRQL are shown in Figure 4. A significant improvement in HRQL was demonstrated for the subscales Respiratory Complaints (7.6 (95% CI 1.9/13.4); *p* = 0.012), Attendant Symptoms and Sleep (9.3 (95% CI 3.1/15.5); *p* = 0.005) and Summary Scale (4.5 (95% CI 0.9/8.0); *p* = 0.016). No significant improvement was observed for the subscales Physical Functioning (4.7 (95% CI −0.7/10.1); *p* = 0.082), Social Relationships (−0.0 (95% CI −6.9/6.8); *p* = 0.991), Anxiety (3.3 (95% CI −3.5/10.1); *p* = 0.012), Psychological Wellbeing (0.2 (95% CI −5.8/6.3); *p* = 0.934) and Social Functioning (6.0 (95% CI −0.1/12.0); *p* = 0.055). According to ESS measurements, moderate effects on daytime sleepiness were observed following the initiation of NIV therapy (Figure 5). Patients without comorbid OSAS showed a mean reduction in ESS of 1.1 ((95% CI −3.8/1.5); *p* = 0.372) after initiation of NIV, whereas patients with comorbid OSAS showed a mean reduction of 2.6 ((95% CI −5.6/0.4); *p* = 0.085). The effects of NIV on RDI in the whole group versus subgroups are shown in Figure 6. The compliance analysis was based on the average duration (h) of device use. Patients with an average of more than 4 h/day (h/d) were classified as compliant. In the subgroup without comorbid OSAS, 28.6% were considered compliant (mean ± SD 5.4 ± 2.7 h/d), whereas in the subgroup with comorbid OSAS, 22.2% (4.7 ± 2.2 h/d) were compliant based on the defined criteria.

The effects of NIV on spirometry, body plethysmography, blood gas parameters and respiratory muscle function diagnostics are displayed in Appendix A.

## 4. Discussion

The present study has demonstrated the effects of NIV initiation on sleep quality in COPD patients following NIV initiation and aimed to provide a deeper understanding of the outcomes associated with NIV therapy.

The main findings can be summarized as follows. Firstly, it was shown that NIV does not decrease sleep quality. In addition, there is a tendency towards an increase in both slow-wave and REM sleep, and, according to ESS measurements, a moderately positive effect on daytime sleepiness, especially in patients with comorbid OSAS, although statistical significance was not reached.

Secondly, there was a strong indication of a significant improvement in RDI. This was particularly true for patients with comorbid OSAS, who showed a massive reduction in RDI at follow-up after 3 months of NIV therapy.

Thirdly, an increase in disease-specific HRQL was demonstrated, even after the short interval of three months. Moreover, it is important to note that in addition to an improved SRI summary scale, significant improvements in the subscales “attended symptoms and sleep” (AS) and “respiratory complaints” (RC) were evident.

Fourthly, it was confirmed that treatment with NIV leads to an improvement in lung function parameters, as demonstrated by increases in vital capacity and FEV_1_.

Finally, this study was able to reproduce the results of previous randomized, controlled trials which concluded that when a maximal reduction in PaCO_2_ is the primary goal, (high-intensity) NIV is effective in reducing continuous transcutaneously measured nocturnal PaCO_2_ [20,21,22].

The present results scientifically address the research question regarding the role of NIV in treating chronic respiratory insufficiency in patients with COPD, which was flagged by the ERS Task Force as a high-priority topic [5]. The present study was able to show that patients with comorbid OSAS especially benefitted from NIV in terms of daytime sleepiness. In addition, NIV therapy was associated with an increase in REM sleep, which was also more pronounced in patients with comorbid OSAS. Moreover, a massive reduction in REM sleep was demonstrated before the initiation of NIV therapy in these patients compared to healthy individuals [23]. Therefore, the positive effects of NIV therapy on both slow-wave and REM sleep may have been responsible for the observed improvement in sleep quality, which was measured by patient-relevant outcome parameters, namely the SRI questionnaire and ESS.

As previous studies have shown, OSAS is a frequent comorbidity in patients with COPD, and this held true for the cohort in the present study (40%) [7]. Considering the frequent occurrence of OSAS in COPD patients, appropriate diagnostic methods such as polysomnography and polygraphy should become integral parts of clinical practice, since the NIV therapy option appears to have an overall beneficial effect on both of these diseases, and does not have a negative effect on sleep quality [6]. Due to structural expansion of outpatient setups, even for the initiation of NIV therapy, diagnostic methods that can be applied outside the hospital setting are advantageous in this setting [24]. Furthermore, an equivalent level of NIV therapy efficacy was observed in patients who were monitored using polygraphy versus those who were monitored using polysomnography during NIV initiation [25].

Other comorbidities known to affect sleep quality, such as diabetes, obesity, cardiovascular disease and depression, are also common comorbidities in COPD patients [26]. However, the associations between these comorbidities and NIV therapy in relation to sleep quality are still unknown and should be investigated in future studies [27].

Regarding the possible negative influence of higher IPAP levels in the form of HI-NIV, a study by Dreher and colleagues was able to show that even higher IPAP levels do not appear to have a negative influence on sleep quality in patients who have already been established on HI-NIV therapy. However, it should be noted that the critical time window for therapy initiation was not taken into account and a selection bias might therefore exist in the rather small collective of 17 patients [28]. Similar results for sleep quality were obtained in a study by Storre et al., in which 10 patients with pre-existing HI-NIV were switched to target tidal volume NIV. This study also showed that changing the ventilation mode did not negatively influence sleep quality, suggesting that this form of ventilation could also be a therapeutic option for selected patient groups [29].

The limitations of this study are as follows. Due to the pandemic situation, not all patients were able to participate in the final follow-up, resulting in a smaller number of cases available for the final analysis. Nevertheless, the use of high-quality diagnostics such as full-body polysomnography with additional transcutaneous PCO_2_ measurement still allowed clinically relevant conclusions to be drawn from the analyses. Regarding sleep quality, the Arousal Index would be a useful additional parameter, which was not evaluated in this study. Further studies should consider this aspect and additionally assess the Arousal Index and its potential change following the initiation of NIV.

In addition, this study only investigated patients after three months of NIV therapy, a period during which adjustments to ventilation are typically still being made, especially in patients who require a longer period of acclimatization to higher inspiratory pressures. Thus, at the time of inclusion in the study, only a proportion of patients had been established on those pressure levels required to achieve the greatest benefit from high-intensity ventilation therapy. Therefore, further studies are needed to determine the long-term effects of HI-NIV on sleep quality in a larger patient population.

## 5. Conclusions

In conclusion, NIV does not have a negative effect on sleep quality in COPD patients. Furthermore, three months of NIV therapy was associated with positive effects on sleep quality, daytime sleepiness, RDI, transcutaneous PCO_2_ and health-related quality of life, while the positive effects on daytime sleepiness and RDI were even stronger in patients with comorbid OSAS.

## Figures and Tables

**Figure 1 jcm-11-05483-f001:**
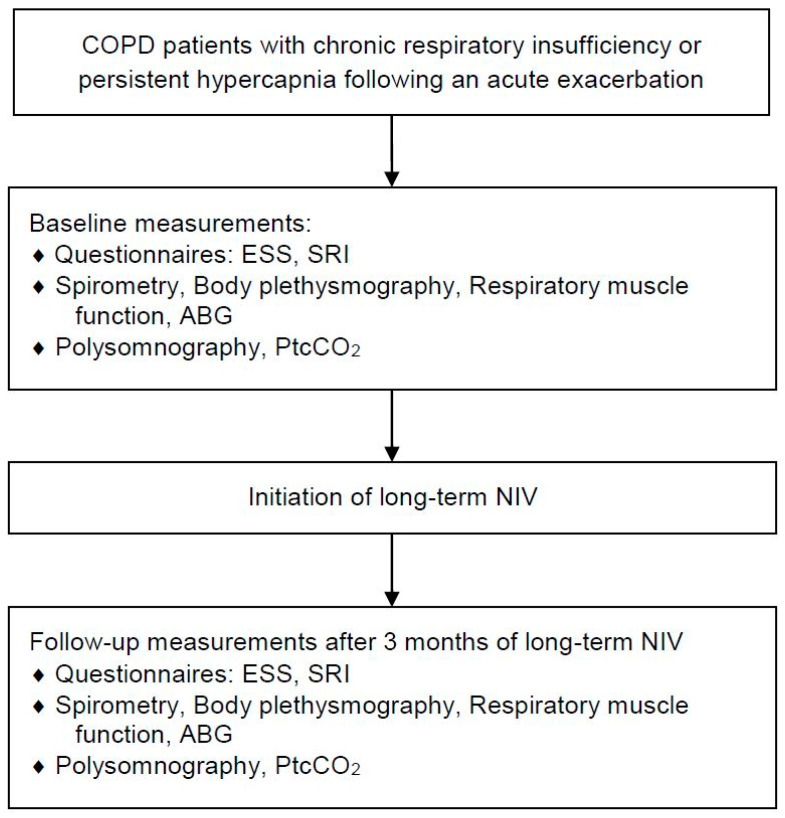
Flow chart of study design and measurements. Abbreviations: NIV: non-invasive ventilation, ESS: Epworth Sleepiness Scale; SRI: Severe Respiratory Insufficiency Questionnaire; ABG: arterial blood gas analysis; PtcCO_2_: transcutaneous PCO_2_.

**Figure 2 jcm-11-05483-f002:**
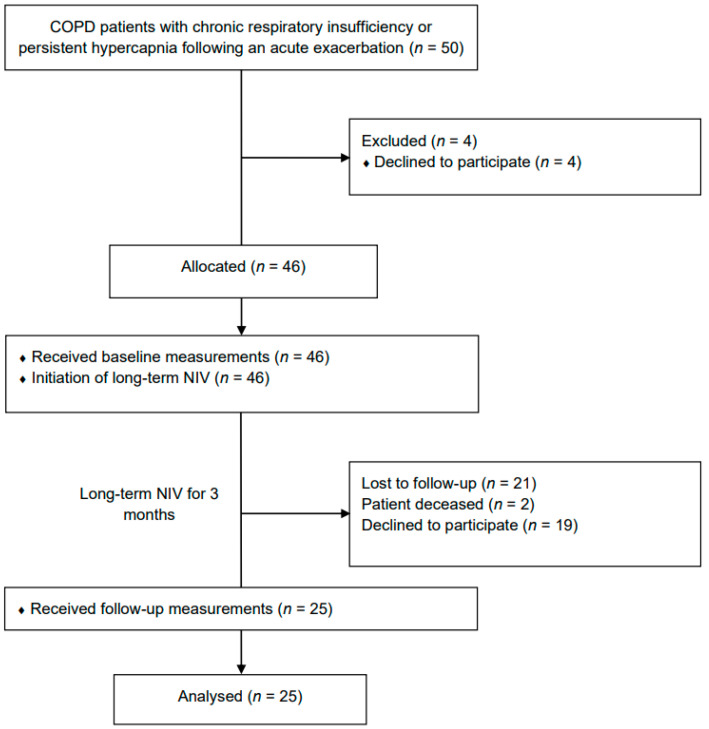
Flow diagram of subject recruitment and data availability. Abbreviations: *n* = number; NIV = non-invasive ventilation.

**Figure 3 jcm-11-05483-f003:**
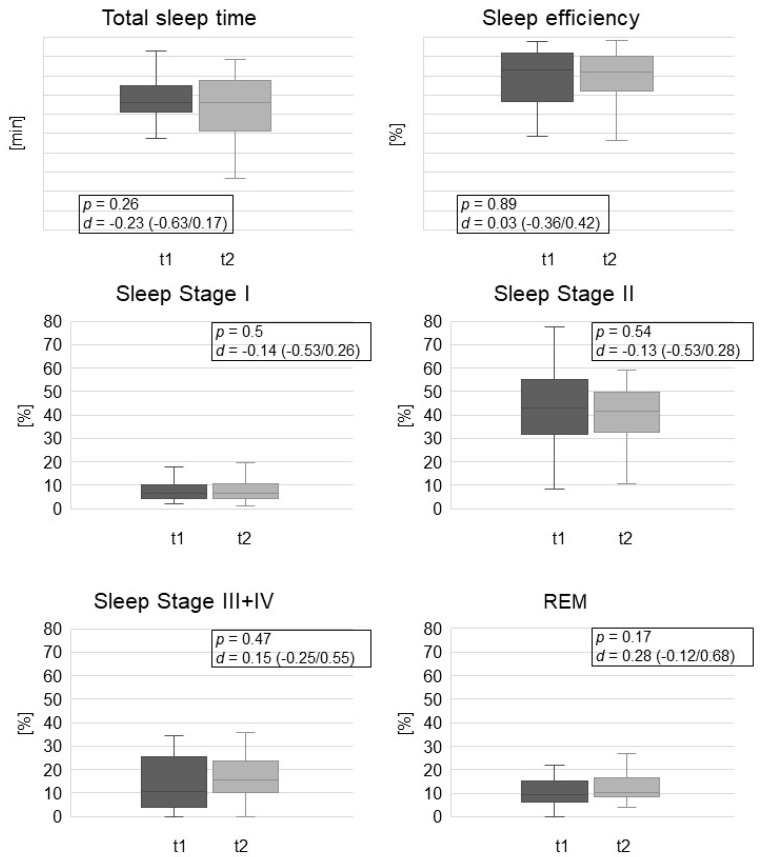
A series of boxplots illustrating the polysomnographic findings prior to NIV (t1) and at 3 months follow-up (t2) in the whole study population (*n* = 25). Notes: *p*-value of 2-tailed *t*-test; data for Cohan’s *d* are displayed with 95% confidence intervals in brackets. Abbreviations: *d*: Cohan’s *d*; m: minutes; REM: rapid eye movement.

**Figure 4 jcm-11-05483-f004:**
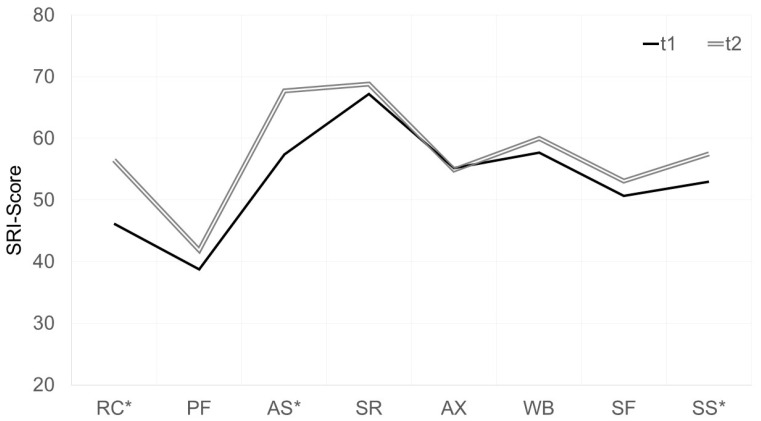
Domains of the Severe Respiratory Insufficiency Questionnaire (SRI). Notes: Higher scores (range, 0–100) indicate a higher HRQL. * Significant mean differences (*p* ≤ 0.05) between t1 (prior non-invasive ventilation) and t2 (following 3 months of NIV). Abbreviations: RC: respiratory complaints; PF: physical functioning; AS: attendant symptoms and sleep; SR: social relationships; AX: anxiety; WB: psychological wellbeing; SF: social functioning; SS: summary scale.

**Figure 5 jcm-11-05483-f005:**
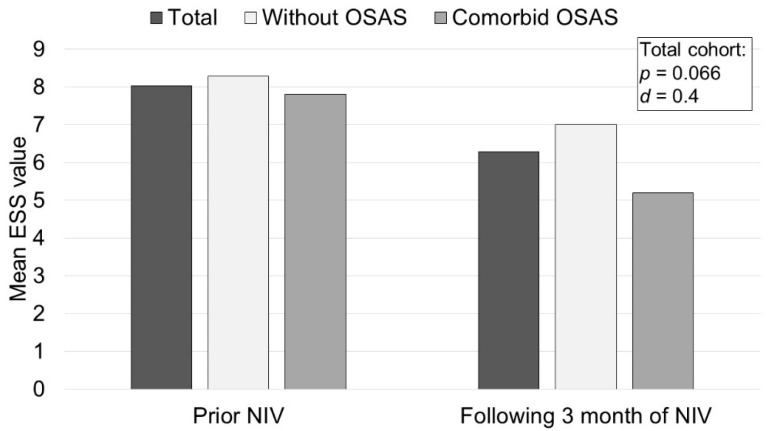
Results of the Epworth Sleepiness Scale (ESS) Questionnaire. Notes: Higher scores (0–24) indicate increased daytime sleepiness. The *p*-value for the 2-tailed *t*-test and Cohan’s *d* for effect size are only reported for the entire cohort (*n* = 25). Abbreviations: OSAS: obstructive sleep apnea syndrome; NIV: non-invasive ventilation; *d*: Cohan’s d.

**Figure 6 jcm-11-05483-f006:**
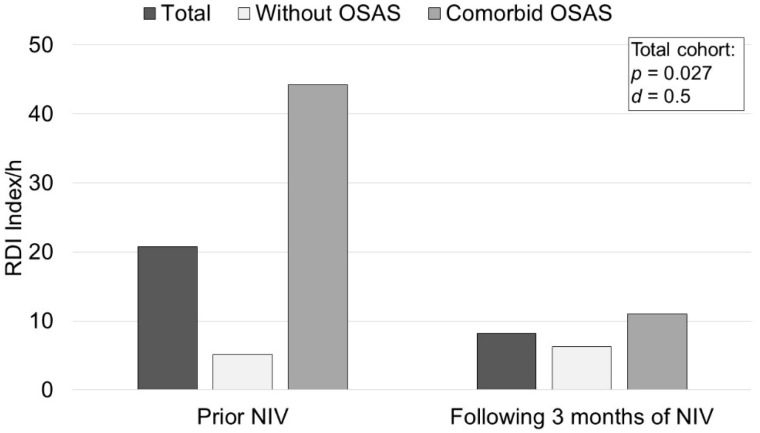
Results of the Respiratory Disturbance Index (RDI). Notes: RDI was used to classify the severity of respiratory events (mild 5–14.9, moderate 15–29.9, and severe >30). The *p*-value for the 2-tailed *t*-test and Cohan’s *d* for effect size are only reported for the entire cohort (*n* = 25). Abbreviations: RDI: index of abnormal breathing events per hour; OSAS: obstructive sleep apnea syndrome; NIV: non-invasive ventilation; *d*: Cohan’s *d*.

**Table 1 jcm-11-05483-t001:** Patient characteristics at baseline.

Characteristics	*n* = 25
Age (years)	66.7 ± 7.4
Female sex (%)	40%
FEV_1_ (L)	0.9 ± 0.4
FEV_1_ after bronchodilation (%pred)	31.8 ± 13.2
FVC (L)	1.6 ± 0.7
FVC (%pred)	47.7 ± 14.9
FEV_1_/FVC ratio (%pred)	50.2 ± 10.0
TLC (L)	6.9 ± 1.8
TLC (%pred)	117.1 ± 29.1
RV (L)	5.1 ± 1.9
RV (%pred)	222.8 ± 84.6
Cumulative smoking dosage (pack years) *	38.1 ± 18.4
BMI (kg/m^2^)	27.7 ± 7.6
pH	7.41 ± 0.03
PaO_2_ (mmHg)	48.4 ± 8.2
PaCO_2_ (mmHg)	48.3 ± 8.0
HCO_3_^−^ (mmol/L)	29.3 ± 3.2
P0.1 (kPa)	0.4 ± 0.2
PI max (kPa)	4.2 ± 2.1
SRI (points)	53.5 ± 14.3
ESS (points)	8.1 ± 4.2

* Twenty patients (80%) were ex-smokers and five patients (20%) were current smokers. Abbreviations: FEV_1_: forced expiratory volume in 1 s; FVC: forced vital capacity; RV: residual volume; TLC: total lung capacity; BMI: body mass index; PaO_2_: partial pressure of arterialized oxygen; PaCO_2_: partial pressure of arterialized carbon dioxide; HCO_3_^−^: arterialized standard hydrogen carbonate; P 0.1: pressure during 0.1 s inspiratory occlusions; PI max: maximal inspiratory pressure; SRI: Severe Respiratory Insufficiency Questionnaire (0–100 points, higher scores indicating better HQRL, minimal clinically relevant difference = 4–5 points); ESS: Epworth Sleepiness Scale (0–24 points, higher scores indicating more daytime sleepiness).

## Data Availability

Data are available upon request from the corresponding author.

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
