# Peer review of "The Impact of Non-Invasive Ventilation on Sleep Quality in COPD Patients"

_jcm, 2022, doi:10.3390/jcm11185483_

Round 1
Reviewer 1 Report
In this paper, the authors aimed to evaluate, through a pilot study, sleep quality and quality of life in very severe COPD patients on nocturnal NIV.
The study is important because it addresses a point that has been poorly explored in COPD patients in NIV. In addition, the use of polysomnography is reaffirmed not only for COPD-overlap patients but also for those without sleep apnea
However, some points need to be clarified:
1) did the authors evaluate the arousal index even unrelated to respiratory disorders before and after the initiation of NIV? It is important to assess and report them because they give an idea of sleep quality.
2) It is not described whether there are patients on oxygen therapy. If so were they receiving oxygen during NIV?
3) NIV initiation criteria should be defined and placed in a separate box.
4) Why did the authors not use the St. George's Respiratory Questionnaire (SGRQ), which is a disease-specific instrument designed to measure the impact on general health, daily living, and perceived well-being in patients with obstructive airway disease? They need to better specify their choice.
Reviewer 2 Report
1. First of all, in line 26 I think there is a mistake regarding the follow up period. please correct
2. in my opinion the conclusion formulation "NIV does not negatively impact on sleep quality" must be reformulated, because does not have scientific soundness. Also, this sentence is reapeated 2 times in the article body. please review.
3. in lines 106-107 - please add AASM and DGSM year of publication and also add this two in the reference list.
4. in line 207-208 - in my opinion you should describe more detailed the compliance parameters.
5. in discussion section you should includ other patients comorbidities, including the ones that can influence the sleep quality, for example cardiovascular diseases.
